# Short-Term Autophagy Preconditioning Upregulates the Expression of COX2 and PGE2 and Alters the Immune Phenotype of Human Adipose-Derived Stem Cells In Vitro

**DOI:** 10.3390/cells11091376

**Published:** 2022-04-19

**Authors:** Rachel M. Wise, Sara Al-Ghadban, Mark A. A. Harrison, Brianne N. Sullivan, Emily R. Monaco, Sarah J. Aleman, Umberto M. Donato, Bruce A. Bunnell

**Affiliations:** 1Neuroscience Program, Tulane Brain Institute, Tulane University School of Science & Engineering, New Orleans, LA 70118, USA; rachel.wise@med.uni-muenchen.de (R.M.W.); mharri26@tulane.edu (M.A.A.H.); bsulliv7@tulane.edu (B.N.S.); emilyrosemonaco@gmail.com (E.R.M.); saleman@tulane.edu (S.J.A.); udonato1@tulane.edu (U.M.D.); 2Center for Stem Cell Research & Regenerative Medicine, Tulane University School of Medicine, New Orleans, LA 70112, USA; sara.al-ghadban@unthsc.edu; 3Department of Microbiology, Immunology and Genetics, University of North Texas Health Science Center, Fort Worth, TX 76107, USA; 4Department of Pharmacology, Tulane University School of Medicine, New Orleans, LA 70112, USA

**Keywords:** adipose tissue-derived stem cells (ASCs), autophagy, rapamycin, 3-methyladenine, immunosuppression, inflammation

## Abstract

Human adipose-derived stem cells (hASCs) are potent modulators of inflammation and promising candidates for the treatment of inflammatory and autoimmune diseases. Strategies to improve hASC survival and immunoregulation are active areas of investigation. Autophagy, a homeostatic and stress-induced degradative pathway, plays a crucial role in hASC paracrine signaling—a primary mechanism of therapeutic action. Therefore, induction of autophagy with rapamycin (Rapa), or inhibition with 3-methyladenine (3-MA), was examined as a preconditioning strategy to enhance therapeutic efficacy. Following preconditioning, both Rapa and 3-MA-treated hASCs demonstrated preservation of stemness, as well as upregulated transcription of cyclooxygenase-2 (COX2) and interleukin-6 (IL-6). Rapa-ASCs further upregulated TNFα-stimulated gene-6 (TSG-6) and interleukin-1 beta (IL-1β), indicating additional enhancement of immunomodulatory potential. Preconditioned cells were then stimulated with the inflammatory cytokine interferon-gamma (IFNγ) and assessed for immunomodulatory factor production. Rapa-pretreated cells, but not 3-MA-pretreated cells, further amplified COX2 and IL-6 transcripts following IFNγ exposure, and both groups upregulated secretion of prostaglandin-E2 (PGE2), the enzymatic product of COX2. These findings suggest that a 4-h Rapa preconditioning strategy may bestow the greatest improvement to hASC expression of cytokines known to promote tissue repair and regeneration and may hold promise for augmenting the therapeutic potential of hASCs for inflammation-driven pathological conditions.

## 1. Introduction

Adipose-derived mesenchymal stem cells (ASCs) have emerged as a promising cell-based therapeutic agent in regenerative medicine and tissue engineering. They possess significant advantages over their bone marrow-derived (BMSC) counterparts due to their ease of harvest, higher stem cell yield, enhanced secretion of immunomodulatory factors and reduced immunogenicity [1,2,3]. As reviewed by Ceccarelli and colleagues, ASCs’ ability to regulate the immune microenvironment gives them immense translational potential in autoimmune, inflammatory, ischemic, and neurodegenerative disease states [4]. However, the lack of standardized cellular processing, donor-to-donor variability, and low ASC survival in the transplant environment remain major obstacles to translational success. ASCs exert their therapeutic benefit predominantly via the production and secretion of immune-modifying proteins and extracellular vesicles (EVs) [3,5,6,7]. In vitro and in vivo studies demonstrated that ASCs amplify their secretory activity following exposure to hypoxic and inflammatory conditions, which are biomimetic of many disease states. Specifically, these conditions serve to enhance ASC production of molecules like transforming growth factor-beta (*TGF-β*), indoleamine-pyrrole 2,3-dioxygenase (*IDO*), prostaglandin E2 (PGE2), and interleukin-10 (*IL-10*) which suppress T cell proliferation and shift macrophages from pro-inflammatory to anti-inflammatory/pro-regenerative phenotypes [5,8,9,10,11,12,13]. Of particular interest is the eicosanoid PGE2, which is synthesized from arachidonic acid by cyclooxygenase-2 (*COX2*), and can modify T cell and macrophage populations in several inflammatory [14,15] and autoimmune [16,17,18,19,20,21] diseases. Although preclinical research supports the efficacy of ASC-based therapeutics, the transition from preclinical to clinical testing has had limited success. The focus has therefore shifted toward optimizing strategies that enhance ASC post-transplant survival and immunosuppression with hopes of bridging this gap [22,23].

Autophagy, derived from the Greek words for “self-eating”, is a multifunctional pathway with key roles in stem cell development, homeostatic maintenance, metabolism, secretory pathways, and stress response [24,25]. Under basal conditions, autophagy serves a housekeeping function by degrading damaged organelles and long-lived proteins via trafficking to the lysosome for enzymatic breakdown. However, stress conditions such as nutrient deprivation, hypoxia, and inflammation trigger signaling cascades that activate autophagic flux to maintain protein synthesis and promote survival [26]. Numerous pharmacological agents can be used to manipulate the autophagy pathway. Two of the most common are rapamycin (Rapa), an immunosuppressive drug that induces autophagy by silencing its negative regulator, mammalian target of rapamycin (mTOR); and 3-methyladenine (3-MA), which inhibits the initiation of autophagy through its blockade of vacuolar protein sorting 34 (VPS34) [27,28]. To date, few investigations have explored the correlation between human ASCs (hASCs) autophagy and immunomodulatory capacity [29,30,31]. In hASCs, Li and colleagues demonstrated that autophagy induction with short-term Rapa exposure suppressed activation of caspase 3 and enhanced overall survival in response to oxygen–glucose deprivation (OGD). In contrast, the inhibition of autophagy proteins with an shRNA approach exacerbated cell death [31]. The authors suggested that Rapa may protect hASCs by enhancing their survival rate after therapeutic administration into hypoxic tissue niches. Another study conducted by Kim and collaborators showed that Rapa preconditioning of hASCs increased both mRNA and protein expression of the immunosuppressive cytokines *IDO*, *IL-10*, and *TGF-β,* suppressed T helper 17 (T_h_17) cells thereby promoting T regulatory cells (T_regs_), and prolonged survival in a mouse model of graft-versus-host-disease (GVHD) [29]. Conversely, Javorka et al. demonstrated that Rapa did not enhance the expression of anti-inflammatory markers like *TSG-6*, *IDO*, and *COX2* in hASCs either at rest or following interferon gamma (IFNγ) stimulation [30]. These contradictory findings suggest that autophagy may play a complicated role in regulating hASC immunomodulatory capacity and may be dependent on undefined variables which must be investigated prior to clinical applications.

In this study, the impact of pharmacological induction and inhibition of autophagy on hASC stemness and immunomodulatory behavior was investigated by using Rapa and 3-MA, respectively. The effects of short-term (4 h) versus long-term (24 h) autophagy preconditioning were also compared to determine the optimal strategy for enhancing production of immune-modifying factors. The data demonstrate that hASCs maintained their stemness regardless of pharmacological agent or duration of exposure. However, expression of pro- and anti-inflammatory mediators differed between autophagy-induced and autophagy-inhibited cells, and also between exposure times, showing that transient versus prolonged manipulation of autophagy has different impacts on hASC physiology. Interestingly, both Rapa and 3-MA-preconditioned hASCs exhibited a time-dependent increase of *COX2* gene expression and secretion of PGE2. Overall, our findings demonstrate that short-term preconditioning of hASCs with Rapa represents a novel strategy for enhancement of their immunomodulatory potential. This may prove beneficial in enhancing the translation of ASC-based therapies from animal models to human patients.

## 2. Materials and Methods

### 2.1. Cells and Cell Culture

Primary human ASCs (hASCs) were purchased from LaCell LLC (New Orleans, LA, USA). Each hASC cell line underwent full characterization individually prior to being pooled together [32,33,34,35,36]. Pooled hASCs from 5 healthy donors were thawed at passage 3 and maintained in complete culture medium (CCM) consisting of minimum essential medium alpha (Cat #: 12561; Gibco, Grand Island, NY, USA) supplemented with 10% heat-inactivated fetal bovine serum (FBS, Cat #: SH30396.03; Thermo Fisher, Waltham, MA, USA), and 1% penicillin-streptomycin (Cat #: 15140122; 10,000 U/mL, Thermo Fisher) in a humidified 5% CO_2_ incubator. Media was changed every 2–3 days until cells achieved 80% confluence then harvested with 0.25% trypsin/1 mM EDTA (Cat #: 25200056; Thermo Fisher) and passaged. For all experiments in this study, cells were used at passage 5. Complete donor information is listed in Table 1.

### 2.2. Western Blot Analysis

hASCs were seeded at a density of 4 × 10^3^ cells/cm^2^. After 72 h, cells were preconditioned with either Rapa (500 nM in DMSO, Cat #: 553211; Millipore Sigma, Burlington, MA, USA), or 3-methyladenine (3-MA; 5 mM in CCM, Cat #: M9281; Millipore Sigma) for 1, 4, 12, 24, and 48 h. After autophagy preconditioning, hASCs were washed once with ice-cold 1X phosphate-buffered saline (PBS) and lysed with RIPA lysis buffer (Cat #: 89900; Thermo Fisher) supplemented with 1X protease inhibitor (Cat #: 1862209; Thermo Fisher) and 1X phosphatase inhibitor (Cat #: 1862495; Thermo Fisher). Protein samples were quantified by using the bicinchoninic acid assay (BCA, Cat #: 23225; Thermo Fisher), and 10 µg of protein lysate was resolved with SDS-PAGE on 4–12% Bis-Tris gels (Cat #: NW04122BOX; Thermo Fisher) in NuPAGE LDS sample buffer (Cat #: NP0007; Thermo Fisher) by using the XCell SureLock Mini-Cell Electrophoresis system (Thermo Fisher). Separated proteins were then transferred onto a PVDF membrane (Cat #: IB401032; Thermo Fisher) by using the iBLOT semi-dry transfer system (Thermo Fisher). The membrane was immediately blocked with 5% non-fat milk in TBS-T (150 mM NaCl, 0.1% Tween 20, 25 mM Tris-HCl, pH 7.6) for 1 h at room temperature (RT). The membrane was then probed with primary antibodies for LC3 (1:1000; Cat #: 27755; Cell Signaling Technologies, Danvers, MA, USA), p62/SQSTM1 (1:2000; Cat #: ab56416; Abcam, Cambridge, UK) and β-*actin* (1:2000; Cat #: 8457S; Cell Signaling Technologies) according to the manufacturer’s protocol. After incubating with HRP-conjugated goat anti-mouse or goat anti-rabbit secondary antibodies (Cat #: 7076P2 or 70745; Cell Signaling Technologies), the membrane was incubated with Clarity Western ECL substrate (Cat #: 1705061; Biorad, Hercules, CA, USA) for 5 min and immediately imaged by using the ImageQuant 4000 Imaging System (GE Healthare, Chicago, IL, USA). Captured images were analyzed by using the ImageJ software (U. S. National Institutes of Health, Bethesda, MD, USA). Expression levels of all proteins were normalized to β-*actin*.

### 2.3. Flow Cytometry

For phenotypic analysis of hASC surface marker expression, cells were blocked with 1% CD16/CD32 in 1X PBS supplemented with 1% bovine serum albumin (BSA) and stained with the following fluorochrome-conjugated primary antibodies at 4 °C for 15 min: CD3 (Cat #: 562406, BD Biosciences, San Jose, CA, USA), CD14 (Cat #: IM2640U, Beckman-Coulter, Brea, CA, USA), CD31 (Cat #: 563651, BD Biosciences), CD45 (Cat #: A71117, Beckman-Coulter), CD73 (Cat #: 550257, BD Biosciences), CD90 (Cat #: 11-0909-42, Thermo Fisher, Waltham, MA, USA), and CD105 (Cat #: 17-1057-42, Thermo Fisher). Stained cells were then fixed with 1% paraformaldehyde (PFA) for 5 min at RT and at least 5000 events were captured by using a Gallios flow cytometer (Beckman Coulter) and analyzed by using Kaluza Analysis 2.1 software (Beckman Coulter).

### 2.4. Colony Forming Unit-Fibroblast (CFU-F) Assay

hASCs were seeded at a density of 250 cells per 10 cm^2^ and allowed to adhere for 24 h. Cells were autophagy preconditioned for 4 h, washed, then cultured for 14 days to allow for colony formation. The medium was changed on day 7 and on day 14 the cells were washed twice with 1X PBS and stained with 3% crystal violet (MilliporeSigma, St. Louis, MO, USA) in methanol for 30 min at RT. The plates were then washed with deionized water until clear and the number of colonies larger than 2 mm in diameter were manually recorded.

### 2.5. RNA Isolation and Quantitative Reverse-Transcription PCR (qRT-PCR)

Total RNA was collected from lysed hASCs, and RNA extraction was performed by using the Qiagen RNeasy Plus mini kit (Cat #: 74136, Qiagen, Germantown, MD, USA). A total of 1 µg of mRNA was then used for cDNA synthesis by using the Applied Bioscience High-Capacity cDNA Reverse Transcription kit (Cat #: 4368814, Thermo Fisher). qRT-PCR was performed by using the SsoAdvanced Universal SYBR Green Supermix (Cat #: 1725271, Bio-Rad, Hercules, CA, USA). Exon-spanning human-specific primers were designed by using the Primer-BLAST online tool. Primer sequences used for qRT-PCR are listed in Table 2. All reactions were performed in duplicate. Analysis was performed by using the 2^−ΔΔCt^ method to calculate the relative fold-change in transcript expression after normalization to the reference gene, β-*actin*.

### 2.6. Enzyme-Linked Immunosorbent Assay (ELISA)

To determine the concentration of secreted PGE2, hASCs were preconditioned with either control, Rapa, or 3-MA-containing media for 4 or 24 h. Following a wash to remove residual autophagy compounds, cells were exposed to human IFNγ (hIFNγ) (5 ng/mL in sterile water; Cat #: PHC4031; Thermo Fisher). After 24 h, the conditioned medium (CM) was collected, centrifuged to remove cellular debris, and stored at −80°C until use. Levels of PGE2 were then measured by using the Prostaglandin E2 Parameter Assay Kit (Cat #: KGE004B; R&D Systems, Minneapolis, MN, USA) according to the manufacturer’s protocol. Briefly, CM was thawed at RT and equal volumes from three independent experiments were pooled together, diluted three-fold, then incubated in the wells of a 96-well pre-coated plate at room temperature. Wells were incubated with a capture antibody, an HRP-conjugated PGE2 competitor, a substrate solution, and finally a stop solution. Absorbance was then read at 450 nm on a Synergy HTX plate reader (BioTek, Winooski, VT, USA). Each sample was standardized to the appropriate negative controls, and the PGE2 concentration was extrapolated from the standard curve.

### 2.7. Statistical Analysis

All data are presented as mean ±SEM of at least three independent experiments and GraphPad PRISM 8 was used to perform all statistical analyses. Results were compared by using one-way analysis of variance (ANOVA) followed by a Tukey’s post-hoc test to analyze the differences between multiple groups. Asterisks (*) denote statistical significance: * *p*< 0.05; ** *p* < 0.01; and *** *p* < 0.001.

## 3. Results

### 3.1. The mTOR Inhibitor Rapamycin Induces, While the PI3K Inhibitor 3-MA Suppresses, Autophagy in hASCs

In this study, hASCs were treated with rapamycin (Rapa-ASCs) for 1, 4, 12, 24 and 48 h, then examined for the expression of key autophagy genes and proteins. Transcriptional analysis of Rapa-ASCs demonstrated significant upregulation of the autophagy genes *ATG7* and *LC3B* following prolonged, but not short-term exposure to Rapa (Appendix A), suggesting induction of autophagy. A Western blot analysis revealed increased protein levels of LC3-II/β-Actin and decreased protein levels of p62/β-Actin, however this failed to reach significance (Appendix A). To examine inhibition of autophagy, hASCs were treated with 3-MA (3MA-ASCs) for 1, 4, 12, and 24 h and then probed for changes in autophagy gene and protein levels. Due to previous reports of autophagy induction with prolonged exposure to 3-MA [37], the longest timepoint (48 h) was not included for this group. The 3MA-ASCs exhibited no change to LC3-II or p62 transcript or protein levels, indicating no induction of autophagosome formation (Appendix A).

### 3.2. Autophagy Preconditioning Does Not Alter hASCs Stem Cell Properties

To characterize the impact of autophagy preconditioning on stem cell characteristics, RapaASCs and 3MA-ASCs were examined for morphology, clonogenicity, and surface marker expression after either short-term (4 h) or long-term (24 h) treatment. Results demonstrated that hASCs maintained their spindle-like morphology with both short-term and long-term exposure to either Rapa or 3-MA (Figure 1A). Autophagy preconditioning did not significantly alter hASCs immunophenotype as measured by flow cytometric analysis of surface proteins (Figure 1B,C). In both Rapa-ASCs and 3MA-ASCs, positive expression of canonical MSC markers CD90 and CD105 ranged from 90–100% across all time points, and expression of CD73 ranged from 29.63% to 45.02% with no significant difference between groups (Figure 1B). Immunophenotype of hASCs was further confirmed by the absence of negative MSC markers including the pan T-cell marker CD3, the monocyte/macrophage marker CD14, the endothelial cell marker CD31, and the broad lymphohematopoietic lineage marker CD45. In both 3MA-ASCs and Rapa-ASCs, expression of these negative markers remained low and was not significantly different from untreated hASCs. Additionally, self-renewal ability of treated ASCs was measured by performing a colony-forming unit-fibroblast assay. Analysis of colony-forming units (CFUs) demonstrated no alteration in self-renewal capacity of Rapa-ASCs (Figure 1D). However, the 3MA-ASCs showed significantly reduced self-renewal capacity (Figure 1E).

### 3.3. Autophagy Preconditioning of hASCs Alters Expression of Both Anti-Inflammatory and Pro-Inflammatory Mediators

To determine whether pharmacologically targeting the autophagy pathway alters the immune-modifying abilities of hASCs, cells were treated for either 4 or 24 h with Rapa or 3-MA and then analyzed with RT-qPCR for expression of common immune mediators. In Rapa-ASCs, 4 h, but not 24 h, treatment produced robust enhancement of *TSG-6* mRNA (Figure 2A). The opposite result was seen in 3MA-ASCs, with only the 24-h group demonstrating enhanced expression of *TGF-β* and, to a lesser extent, *TSG-6* (*p* = 0.073). Further analysis revealed similar time-dependent differences in expression of pro-inflammatory mediators (Figure 2B). In both Rapa-ASCs and 3MA-ASCs, *IL-6* was upregulated after 4 h, but not 24 h. Moreover, Rapa-ASCs, but not 3MA-ASCs, upregulated *IL-1*β after 4-h treatment. Interestingly, both 4-h Rapa-ASCs and 3MA-ASCs robustly upregulate *COX2* transcription relative to control cells. This effect was not mirrored in secreted protein, as ELISA analysis of conditioned medium revealed elevation of the *COX2* metabolite, PGE2, in 24-h treated 3MA-ASCs only (Figure 2C).

### 3.4. Autophagy Preconditioning of hASCs Alters Response to Pro-Inflammatory Stimulation with hIFNγ

To investigate the effects of autophagy preconditioning on hASC immunomodulatory response to signals they may encounter following administration into an inflammatory microenvironment, cells were pre-treated for either 4 or 24 h followed by 24 h stimulation with the classic pro-inflammatory cytokine hIFNγ. Our findings demonstrate that, in comparison to unstimulated and non-pretreated control ASCs, 4 h Rapa-preconditioning resulted in significant elevation of *TGF-β, COX2*, and *IL-6*, whereas 24 h Rapa-preconditioning resulted in upregulation of *TGF-β, IDO, TSG-6,* and *COX2,* as well as the inflammatory cytokines *IL-6* and *IL-1β*. We also found that 4 h 3MA-preconditioning resulted in upregulation of *IDO, COX2, IL-6*, and *IL-1β*, whereas24 h 3MA-preconditioning only increased expression of *COX2* transcripts (Figure 3A,B). This contrasted with our non-preconditioned control ASCs, which in response to hIFNγ stimulation only displayed significant upregulation of *IDO* and *COX2* transcripts when compared to non- IFNγ-stimulated ASCs. Perhaps most interesting, we demonstrated that secretion of PGE2 was upregulated by both 4 h Rapa and 4 h 3MA, which was significantly higher than both unstimulated control and IFNγ-stimulated control cells, as well as both 24 h preconditioned groups (Figure 3C). Taken together, these findings show that although transcriptional activity is altered based on unique temporal and compound-dependent patterns, the robust upregulation of PGE2 release in both acute Rapa and 3-MA preconditioning strategies suggest the involvement of an autophagy-independent mechanism that is currently unknown.

## 4. Discussion

ASCs have shown substantial preclinical promise as therapeutic tools in inflammatory, autoimmune, and neurodegenerative diseases [38]. However, the successful application of ASCs in human clinical trials has faced many challenges, including variable clinical efficacy outcomes, low post-transplant viability, and immunosuppressive potency [39]. Preconditioning strategies to improve immunomodulatory potency of MSCs for various anti-inflammatory and regenerative medicine applications is an active focus of the stem cell research field. Rapamycin has emerged as a promising candidate compound, and Rapa-treated MSCs from adipose [29,40], bone marrow [41], and umbilical cord [42] have been examined in preclinical models. In a 2016 report by Kim and colleagues, hASCs were treated with Rapa for 48 h prior to administration in a GvHD mouse model. Authors demonstrate elevated expression of IL-10, IDO, and TGF-β, and correlated this with enhanced production of anti-inflammatory cytokines, modulation of the T cell repertoire, and prevention of GvHD development [29]. However, recent reports have revealed temporally distinct actions of Rapa on mTORC1 and mTORC2 [43], and this has been correlated to dynamic effects on MSC immunomodulatory capacity. Indeed, a wide range of dosage and exposure time has yielded varying success and highlights the importance of defining the unique immunoregulatory consequences of these novel preconditioning strategies depending on experimental conditions (i.e., dose and exposure time), cell type, and therapeutic application. For example, short-term (2 h) treatment of rat BMSCs improved survival and repair of damaged myocardium following transplant into an ischemia/reperfusion model [41]. However, in an animal model of multiple sclerosis, short-term (4 h) preconditioning of human ASCs did not yield any therapeutic benefit and was in fact correlated with worsened disease measures [40]. Thus, to improve translational potential, it is critically important to define the dynamic immunomodulatory response of MSCs and fine-tune these strategies to each pathological situation.

In the present study, the impacts of compounds known to either induce or inhibit autophagy on hASC immunomodulatory potential was assessed. Our data showed that 24 h Rapa treatment increased the expression of the autophagy genes *ATG7* and *LC3B*, whereas the expression of LC3-II protein was upregulated, albeit slightly, suggestive of autophagic induction. Conversely, 3-MA, a class I and class III PI3K inhibitor, obstructed autophagy initiation in hASCs. 3MA-ASCs displayed no significant change to autophagy transcripts or protein levels. Both Rapa-ASCs and 3MA-ASCs maintain characteristic stem cell properties, including plastic adherence, fibroblast-like morphology, and immunophenotypic profile, as determined by flow cytometry. However, because the immunomodulatory strength of ASCs is due to their paracrine activity rather than proliferation and engraftment, this likely has little impact on therapeutic potential. These results denote the retention of basic MSC identifiers in autophagy preconditioned hASCs.

To determine the role of autophagy preconditioning in hASC immunosuppressive potential, both the transcription and secretion of known ASC-derived pro- and anti-inflammatory mediators was examined. Among these, the pleiotropic signaling molecule PGE2 was of particular interest due to its established role in MSC-based therapies for inflammatory and autoimmune diseases including MS [18,19], sepsis [14], inflammatory bowel disease (IBD) [20,21], ischemia-reperfusion injury [15], and arthritis [16,17]. hASC-derived PGE2 exerts its immunomodulatory functions through the inhibition of T cell activation, proliferation and production of pro-inflammatory cytokines [17,44,45,46,47], the generation of IL-10-producing T regulatory cells (Tregs) [17], and the promotion of M2 macrophage polarization [21,48]. PGE2 is synthesized from arachidonic acid by the enzyme COX2 and has been proposed as an important mechanism contributing to the immunoregulatory actions of MSCs. In a mouse model of sepsis, concurrent IV administration of BMSCs pre-stimulated with LPS resulted in prolonged survival, which was correlated with elevated PGE2 secretion and modulation of host macrophage populations [14]. PGE2 has also been shown to be critical for MSCs’ ability to inhibit the proliferation of natural killer (NK) cells [49], the maturation of dendritic cells (DC) from monocytes [45,50], and the proliferation of PHA-stimulated T cells [45].

In this study, 4 h preconditioned Rapa-ASCs and 3MA-ASCs both exhibited robust upregulation of *COX2* gene expression. This induction was not seen in secreted PGE2, the protein substrate of COX2, with autophagy preconditioning alone in either 4 h treatment group; however, PEG2 was significantly increased in ASCs treated with 3-MA for 24 h. Following exposure to hIFNγ, elevated *COX2* transcription and PGE2 secretion was seen in both Rapa-ASCs and 3MA-ASCs with 4 h, but not 24 h, preconditioning. This indicates that short-term autophagy preconditioning may “prime” hASCs to respond more rapidly and more robustly to an environment mimicking the inflammatory state of autoimmune or neurodegenerative diseases [51,52,53,54].

These unique temporal dynamics of COX2 and PGE2 have previously been shown in BMSCs, and have been suggested to result from activation of the Akt/glycogen synthase kinase 3 Beta (GSK-3β) pathway rather than autophagy [55,56]. mTOR, a serine/threonine kinase which forms the catalytic subunit of the two distinct complexes mTORC1 and mTORC2, regulates a spectrum of cellular processes including cell growth, autophagy, cytoskeletal remodeling, proteostasis, and metabolism (elegantly reviewed here [57]). Rapamycin, an FDA-approved immunosuppressant, rapidly inhibits mTORC1 activity through interaction with the FRB (FKBP12/rapamycin-binding) domain [58]. Under physiological conditions mTORC1 substrates negatively regulate mTORC2 kinase activity [59]. With Rapa inhibition of mTORC1, mTORC2 kinase activity is disinhibited, leading to phosphorylation of Akt at Ser473 and subsequently disrupting the glycogen synthase kinase 3 beta (GSK3B)-dependent blockade of the COX2 promotor region, thereby activating COX2 transcription [37,55,60]. With prolonged Rapa exposure mTORC2 is also inhibited, possibly explaining the rapid yet transient COX2 upregulation [43]. The temporally defined actions of 3-MA may also explain the observed transcriptional changes. 3-MA persistently inhibits class I PI3K whereas its inhibition of the class III PI3K, VPS34, is transient [37]. This was correlated with transient inhibition of autophagy and upregulated COX2 levels in human BMSCs with short-term exposure, but induction of autophagy and suppression of COX2 with long-term treatment. Thus, it may be that the selective inhibition of mTORC1 kinase activity, independent of the effects on autophagy, is responsible for the altered immunophenotype of ASCs.

Wang and colleagues demonstrated that short-term, but not long-term, inhibition of mTORC1 with Rapa resulted in elevated *COX2* gene expression, PGE2 secretion, and inhibition of proliferation in PBMCs. However, ASCs represent a more abundant and readily available source of stem cells compared to BMSCs, and possess higher immunomodulatory potential [3]. Thus, the present study investigated whether ASCs demonstrate a similar temporally distinct response to Rapa treatment to determine if this therapeutic strategy may be extended to a novel cell type. Due to the observation that both Rapa and 3-MA elicited some shared immunoregulatory responses in hASCs, it may be that the effects of these preconditioning strategies are independent of autophagy, which was suggested by Chinnadurai and collaborators in 2015 [61]. However, in this study hBMSC were treated with 3-MA for more than 48 h, and based on the temporal dynamics of class I versus class III PI3K inhibition it is possible that the inhibitory effect of 3-MA had weakened or stopped. Our results, consistent with observations in BMSCs, highlight the importance of fully understanding the temporal aspects of autophagy preconditioning to optimize therapeutic potential.

Evidence from two studies indicates that PGE2 is only partially responsible for the immunosuppressive actions of hASC function as inhibition of PGE2 or its receptors does not fully abolish immunosuppressive capacity [46,62]. Other anti-inflammatory mediators, including *IDO*, *TSG-6*, and *TGF-β1*, significantly contribute to hASCs’ suppression of innate and adaptive immune cells. IDO, an enzyme that breaks down the essential amino acid tryptophan, promotes polarization of M2b/c macrophages, induces proliferation and *IL-10* production of Tregs, inhibits proliferation of Th cells, and suppresses the cytolytic activity of natural killer (NK) cells [1,5,8,62,63,64]. *TSG-6* plays a pivotal role in M2 macrophage polarization and reduction of inflammation in both colitis and acute pancreatitis mouse models [65,66]. Moreover, TSG-6 has displayed autocrine activity which maintains stemness and downregulates IL-6 production in mouse BMSCs [67]. *TGF-*β1 is necessary for the suppression of dendritic cell maturation and subsequent promotion of FOXP3^+^ Tregs [68,69]. Additionally, Rapa-treated hBMSCs showed upregulation of *TGF-β1*, which was indispensable for the inhibition of CD4^+^ T cell proliferation [70]. The present study showed that *IDO* was expressed at low levels in resting hASCs and was unaffected by autophagy preconditioning itself. Upon hIFNγ stimulation, both 4 h 3MA-ASCs and 24 h Rapa-ASCs amplified *IDO* expression compared to control stimulated cells. *TSG-6* expression was not significantly changed in resting or hIFNγ-stimulated 3MA-ASCs but was highly upregulated in 4 h preconditioned Rapa-ASCs. Further upregulation of transcription was seen in 24 h preconditioned, hIFNγ-stimulated Rapa-ASCs. Finally, only 24 h preconditioned 3MA-ASCs exhibited upregulation of *TGF-β1* transcripts. After hIFNγ stimulation, both 4 h and 24 h Rapa-ASCs elevated *TGF-β1* gene expression. Overall, Rapa-ASCs demonstrate more upregulation of anti-inflammatory genes than 3MA-ASCs, suggesting Rapa preconditioning may be a more promising strategy for augmentation of hASC immunomodulatory ability.

The pleiotropic cytokine IL-6 can act as a pro-inflammatory or pro-regenerative agent dependent on its actions on its soluble and membrane-bound receptors, respectively [71,72]. In the context of hASCs, IL-6 production has been linked to repression of Th17 cells, induction of IL-10-producing Tregs, and enhanced recruitment and anti-inflammatory polarization of macrophages [73,74,75,76]. In our study, both 4 h Rapa-ASCs and 3MA-ASCs showed a significant increase in *IL-6* transcript levels. Following pro-inflammatory activation with hIFNγ, both 4- and 24 h Rapa-ASCs, and 4 h 3MA-ASCs showed enhanced *IL-6* transcription. Finally, we assessed the expression of *IL-1*β, the potent pro-inflammatory cytokine which negatively correlates with hASCs’ ability to suppress CD4^+^ T helper, CD8^+^ T effector, and NK cell proliferation [77]. In our study, 24 h Rapa preconditioning and 4 h 3-MA preconditioning followed by stimulation with hIFNγ resulted in additional upregulation of *IL-1*β. Taken together, the upregulated *IL-6* and unaltered *IL-1*β expression levels suggest that 4 h Rapa-ASCs may be the most effective autophagy preconditioning strategy.

In summary, these data demonstrate that preconditioning with both the autophagy-inducer Rapa and the autophagy-inhibitor 3-MA had distinct effects on the expression of immunomodulatory factors in hASCs based on the duration of treatment and presence or absence of pro-inflammatory stimuli. The results suggest that the conflicting data in the autophagy literature may be due, at least in part, to varied treatment times, doses, and the unique inflammatory milieu. We also demonstrated that both compounds had similar impact on secretory activity of ASCs in response to inflammatory activation, which strongly suggests a shared molecular mechanism that is independent of their actions on autophagy. Based on our findings, we propose that a short-term preconditioning with Rapa may bestow the most robust enrichment of hASC immunomodulatory potential due to the preservation of stemness qualities, the enhancement of the *COX2*/PGE2 pathway, and the increase of the anti-inflammatory cytokines *TGF-β* and *IL-6* without concomitant elevation of pro-inflammatory *IL-1β.* This is consistent with previous investigations which suggest that short-term Rapa treatment in MSCs elicits greater improvement of immunomodulatory function, possibly resulting from its differential inhibition of MTORC1 and MTORC2 [41,55,78]. On the other hand, 3MA-ASCs and Rapa-ASCs both upregulate PGE2 secretion when activated with hIFNγ, indicating that autophagy inhibition with 3-MA may also have therapeutic benefit in certain disease contexts, as was proposed by Dang and colleagues in the experimental autoimmune encephalomyelitis (EAE) mouse model [19]. Future investigations involving co-cultures with innate and adaptive immune cells and administration to preclinical disease models will be needed to determine the influence of these novel preconditioning strategies on the immunosuppressive function of hASCs. These approaches may reveal whether our in vitro findings translate to enhanced immunomodulatory function and therapeutic efficacy in autoimmune, inflammatory, and neurodegenerative diseases in vivo.

## 5. Conclusions

Adipose-derived stem cells (ASCs) secrete a variety of anti-inflammatory molecules that shift immune cell activity away from driving inflammation and toward tissue regeneration and repair, showing immense therapeutic promise in several inflammatory, autoimmune, and neurodegenerative diseases. However, their efficacy is limited by poor survival and secretory activity after administration, representing a significant hurdle to success in clinical trials. To overcome these hurdles, a novel preconditioning strategy is proposed that targets a cellular pathway involved in stress response, survival, and secretory activity. Drawing on evidence from bone marrow-derived stem cells (BMSCs), we find that compounds commonly used to modify the autophagy pathway result in “primed” ASCs capable of producing higher levels of immunomodulatory genes and proteins. We demonstrate that PGE2, a key contributing factor to stem cell therapeutic efficacy in multiple sclerosis (MS), sepsis, inflammatory bowel disease (IBD), and arthritis, was highly upregulated in ASCs preconditioned with either short-term Rapa or 3-MA, indicating that the immunomodulatory ffects of these compounds may, in fact, derive from mechanisms of action beyond their impact on autophagy. We propose that this represents a promising strategy for enhancing the therapeutic potential and possibly the translational success of ASCs for inflammation-driven disease states that warrants further investigation and delineation of mechanisms involved.

## Figures and Tables

**Figure 1 cells-11-01376-f001:**
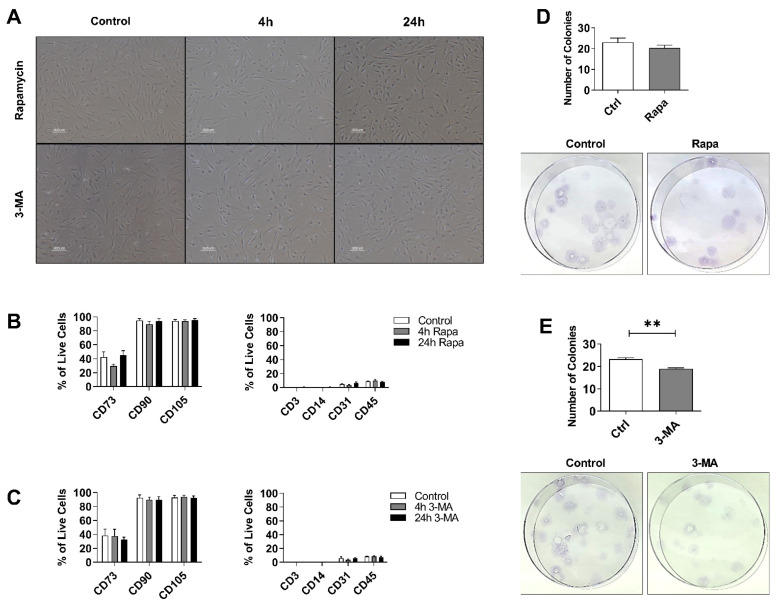
Autophagy preconditioning in hASCs does not alter self-renewal capacity or surface expression of MSC markers. (**A**) Representative images of hASCs morphology in culture after both short= and long-term exposure to autophagy preconditioning agents. Scale bar = 300 µm. Immunophenotype of Rapa-ASCs (**B**) and 3MA-ASCs (**C**) by flow cytometric analysis of surface proteins. Data are presented as mean ± SEM of 3 independent experiments. Quantifaction and images of colony-forming units by Rapa-ASCs (**D**) and 3MA-ASCs (**E**). Data are presented as mean ± SEM of 4 independent experiments. All data comparing 3 groups are analyzed by using one-way analysis of variance (ANOVA) with Tukey’s post-hoc multiple comparisons, and CFU-F data are analyzed with unpaired student’s *t*-test. Abbreviations: Rapa, Rapamycin; 3-MA, 3-methyladenine; CFU-F, colony forming units-fibroblasts. ** *p* < 0.01.

**Figure 2 cells-11-01376-f002:**
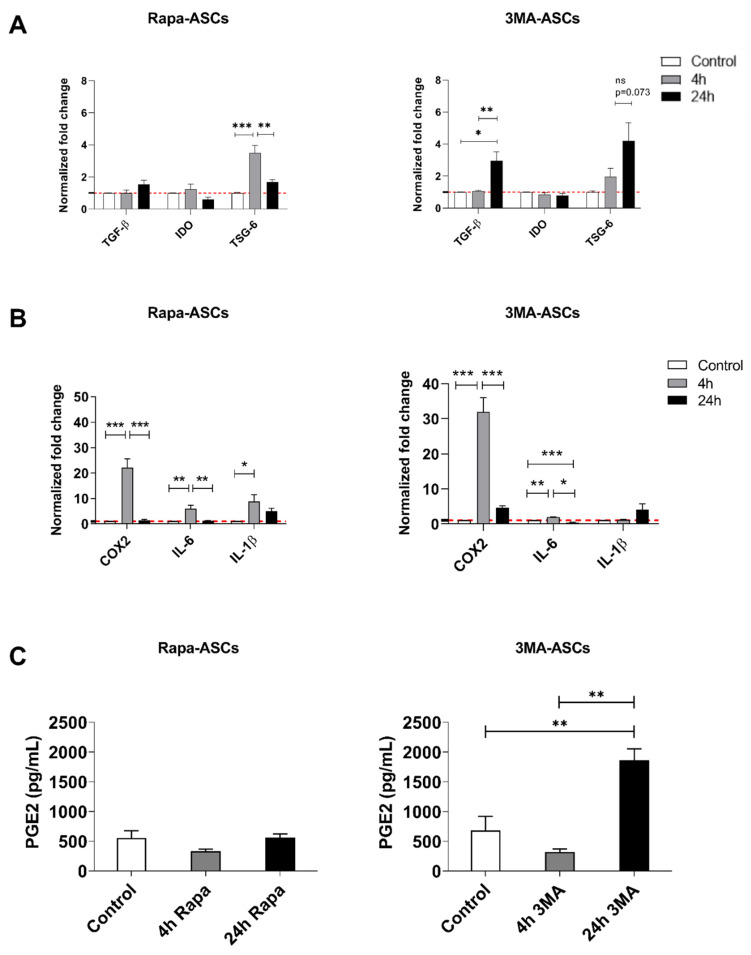
Autophagy preconditioning in hASCs alters expression of both anti-inflammatory and pro-inflammatory mediators. Rapa-ASCs (left) and 3MA-ASCs (right) were treated for 4 or 24 h, then relative expression levels of anti-inflammatory (**A**) and pro-inflammatory (**B**) genes were measured via RT-qPCR by using the ΔΔCt method. Data are presented as mean relative fold-change ± SEM of 4 independent experiments. (**C**) Secreted PGE2 levels were measured via ELISA in 24 h CM from control and autophagy preconditioned hASCs. Data are presented as means ± SEM of 3 independent experiments. Statistical analysis was performed by using one-way analysis of variance (ANOVA), and differences between the means are indicated with * *p* < 0.05, ** *p* < 0.01, *** *p* < 0.001. Abbreviations: *TGF-**β*, transforming growth factor-beta; *IDO*, indoleamine 2,3-dioxygenase; *TSG-6*, TNF Alpha Induced Protein 6, *COX2*, cyclooxygenase 2; *IL-6*, interleukin-6; *IL-1**β*, interleukin-1 beta; PGE2, prostaglandin E2.

**Figure 3 cells-11-01376-f003:**
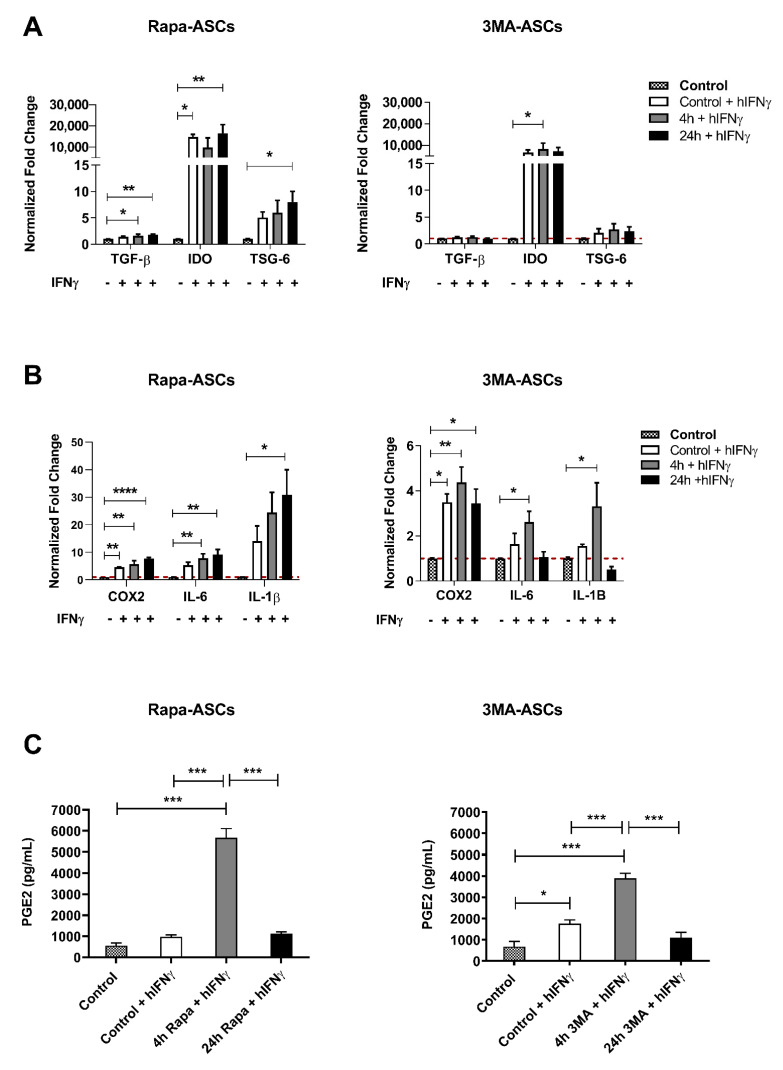
Autophagy preconditioning in hASCs alters their transcriptional and secretory response to pro-inflammatory stimulation with hIFNγ. Rapa-ASCs (left) and 3MA-ASCs (right) were preconditioned for 4 or 24 h with their respective autophagy compounds, then stimulated with hIFNγ (5 ng/mL) for 24 h. Following stimulation, anti-inflammatory (**A**) and pro-inflammatory (**B**) genes were measured by using RT-qPCR. Data are presented as means ± SEM of 4 independent experiments and normalized to controls that were not treated with hIFNγ. (**C**) Secreted PGE2 levels were measured in 24 h CM from autophagy-preconditioned and hIFNγ-stimulated hASCs. Data are presented as means ± SEM of 3 independent experiments. All statistical significance is determined by using one-way analysis of variance (ANOVA). Statistical differences between the means are indicated with * *p* < 0.05, ** *p* < 0.01, *** *p* < 0.001, **** *p* < 0.0001. Abbreviations: TGF-β, transforming growth factor-beta; IDO, indoleamine 2,3-dioxygenase; TSG-6, TNF Alpha Induced Protein 6, COX2, cyclooxygenase 2; IL-6, IL-1β, interleukin-6; interleukin-1 beta; PGE2, prostaglandin E2.

**Table 1 cells-11-01376-t001:** Donor Demographics.

Donor	Age	BMI
1	34	20.34
2	40	21.18
3	39	23.4
4	25	22.0
5	40	21.19
Average ± SD	36.5 ± 2.87	21.62 ± 0.52

**Table 2 cells-11-01376-t002:** Primer Sequences.

Gene	Forward (5′-3′)	Reverse (5′-3′)
*Beta-actin*	ACGTTGCTATCCAGGCTGTGCTAT	TTAATGTCACGCACGATTTCCCGC
*ATG7*	ATGATCCCTGTAACTTAGCCCA	CACGGAAGCAAACAACTTCAAC
*LC3B*	AAGGCGCTTACAGCTCAATG	CTGGGAGGCATAGACCATGT
*P62*	GCACCCCAATGTGATCTGC	CGCTACACAAGTCGTAGTCTGG
*TGF-β*	CAGTCACCATAGCAACACTC	CCTGGCCTGAACTACTATCT
*IDO*	TCTCATTTCGTGATGGAGACTGC	GTGTCCCGTTCTTGCATTTGC
*TSG-6*	AGAATTTGTGAGCAGCCCCT	GGCTGCTCGTTCAAGCCATA
*IL-1β*	CATGGGATAACGAGGCTTATG	CCACTTGTTGCTCCATATCC
*IL-6*	CCTTCCAAAGATGGCTGAAA	TGGCTTGTTCCTCACTACT
*COX2*	TTGCTGGCAGGGTTGCTGGTGGTA	CATCTGCCTGCTCTGGTCAATCGAA

## Data Availability

The original contributions presented in the study are included in the article/supplementary material, further inquiries can be directed to the corresponding author/s.

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
