# Peer review of "Short-Term Autophagy Preconditioning Upregulates the Expression of COX2 and PGE2 and Alters the Immune Phenotype of Human Adipose-Derived Stem Cells In Vitro"

_cells, 2022, doi:10.3390/cells11091376_

Round 1

Reviewer 1 Report

In this study, the authors present a strategy to upregulate the immunomodulatory capacity of adipose-derived stem cells (ASC) through the induction of autophagic pathway by rapamycin, the mTOR inhibitor. ASC cells exposed to mTOR inhibitor show an enhanced immunomodulatory potential, a stable stemness phenotype, the increase of the COX2/PGE2 pathway, and of anti-inflammatory cytokines, which makes them a promising therapeutic intervention in the treatment of several diseases such as inflammatory and neurodegenerative ones. To overcome the well-known issues regarding the clinical use of ASCs (a poor survival and poor secretory activity after in vivo administration), the in vitro preconditioning strategies have been developed to improve both the survival and immune response of ASCs to stressful stimuli.

Overall methods are fit for the target of the work, references list, even if it does not report all recent articles, is properly explaining the state of the art of this topic. However, some modifications should be added to improve the quality of the manuscript.

Major issues:

  • The manuscript is appropriate to the purposes of Cells, but I have to express my concern about the originality of the topic. As far as I know, Kim et al. (2016 - DOI 10.1186/s13287-015-0197-8) published data relative to the autophagy induction in adipose tissue-derived human stem cells by rapamycin treatment able to enhance their immunoregulatory properties.

Also, the manuscript by Wise R.M. and colleagues is very similar to another one published by Wang et al. in 2017 (DOI 10.1186/s13287-017-0744-6) (cited ref n.53) in which it was demonstrated that inhibition of mTOR signalling using short-term rapamycin exposure enhanced the immunosuppressive functions of MSCs from bone marrow through a paracrine mechanism involving the COX-2 upregulation in the presence of IFN-γ and downstream PGE2. The Authors concluded that rapamycin could be a novel approach to improve the MSC-based therapeutic effect. To overcome this issue and to improve the quality of this manuscript, I suggest better emphasizing the originality of the work.

  • The short-term autophagy preconditioning led to a strong upregulation of COX-2 gene expression. In bone marrow MSC it has been demonstrated that COX-1 expression was not affected by rapamycin (Wang et al. 2017 DOI 10.1186/s13287-017-0744-6). Does the COX-1 expression, which is constitutively expressed in most tissues and cells, is affected by rapamycin exposure?

Minor points:

- Statistical analysis: Data from independent experiments are presented as means ± standard deviations (SD). I believe it would be more accurate to use the standard error mean (SEM) instead of standard deviation.

-Lines 117 and 170: Maybe I’m wrong, but I can’t see the Tables? Are they inserted in the supplementary materials or somewhere else?

  • Line 142: It is not necessary or useful to insert the link of the image acquisition program.

- In line 199, authors presented the LC3-II western blot (WB) (Fig S1A) and reported this sentence “Western blot analysis revealed increased protein levels of LC3-II/β-Actin and decreased protein levels of p62/β-Actin, indicative of autophagy initiation (Fig. S1A, B)” but, the increase in the intensity of the LC3-II band is not absolutely evident. Moreover, the densitometric analysis, considering the SD values, shows no significant differences or a time-dependent growing trend. Can you replace the WB image of LC3-II? Moreover, do high rapamycin treatment times induce significant changes in p62 levels? I believe that these slight alterations of LC3-II and p62 expression are not to be considered to unmistakably confirm the autophagy induction.

- Supplementary Figures: Transcriptional analysis of ATG7 shall be included in the methods. Please, insert the legends of the supplementary figures.

I really do not understand the choice of treatment times. Rapamycin is used for 1, 4, 12, 24, and 48 hours but not the 3MA (1, 4, 12, and 24 hours). The Authors should explain the reason.

-Line 224: Correct (Fig. 1C) with (Fig. 1D).

-Lines 224-227: I suggest inserting images of the colonies.

- Figure 2: The COX-2 and relative PGE2 levels are presented. Is the COX-2 expression in the control cells (white square) totally absent? If so, how the evident level of PGE2 can be explained? Please justify.

-Line 289: Please insert   ****p<0.0001

Author Response

We would like to thank all reviewers for their knowledgeable critique and insightful suggestions on how to improve our work entitled “Short-term Rapamycin Preconditioning Diminishes Therapeutic Efficacy of Human Adipose-Derived Stem Cells in a Murine Model of Multiple Sclerosis.” We have addressed all reviewer comments and concerns, and we feel they have improved the revised manuscript. All additional references have been incorporated into the manuscript. The methods have been clarified, and the Abstract, Introduction and Discussion sections have been strengthened due to the integrated edits. We have addressed the comments specific to each reviewer in the following sections.

REVIEWER 1

In this study, the authors present a strategy to upregulate the immunomodulatory capacity of adipose-derived stem cells (ASC) through the induction of autophagic pathway by rapamycin, the mTOR inhibitor. ASC cells exposed to mTOR inhibitor show an enhanced immunomodulatory potential, a stable stemness phenotype, the increase of the COX2/PGE2 pathway, and anti-inflammatory cytokines, which makes them a promising therapeutic intervention in the treatment of several diseases such as inflammatory and neurodegenerative ones. To overcome the well-known issues regarding the clinical use of ASCs (a poor survival and poor secretory activity after in vivo administration), the in vitro preconditioning strategies have been developed to improve both the survival and immune response of ASCs to stressful stimuli.

Overall methods are fit for the target of the work, references list, even if it does not report all recent articles, is properly explaining the state of the art of this topic. However, some modifications should be added to improve the quality of the manuscript.

Major issues:

  1. The manuscript is appropriate to the purposes of Cells, but I have to express my concern about the originality of the topic. As far as I know, Kim et al. (2016 - DOI 10.1186/s13287-015-0197-8) published data relative to the autophagy induction in adipose tissue-derived human stem cells by rapamycin treatment able to enhance their immunoregulatory properties.

Response: We appreciate this comment and agree that further emphasizing the novelty of our study will strengthen its impact. Thus, we have added the following to the Discussion:

Preconditioning strategies to improve the immunomodulatory potency of MSCs for various anti-inflammatory and regenerative medicine applications is an active focus of the stem cell research field. Rapamycin has emerged as a promising candidate compound, and Rapa-treated MSCs from adipose8,9, bone marrow10, and umbilical cord11 have been examined in preclinical models. For example, in a 2016 report by Kim and colleagues, hASCs were treated with Rapamycin for 48 hours before administration in a GvHD mouse model. Authors demonstrate elevated expression of IL-10, IDO, and TGF-β and correlated this with enhanced production of anti-inflammatory cytokines, modulation of the T cell repertoire, and prevention of GvHD development1. However, recent reports have revealed temporally-distinct actions of Rapamycin on mTORC1 and mTORC22; this has been correlated to dynamic effects on MSC immunomodulatory capacity. Indeed, a wide range of dosage and exposure time has yielded varying success and highlights the importance of defining the unique immunoregulatory consequences of these novel preconditioning strategies depending on experimental conditions (i.e. dose and exposure time), cell type, and therapeutic application. For example, short-term (2 hour) treatment of rat BMSCs improved survival and repair of damaged myocardium following transplant into an ischemia/reperfusion model10. However, in an animal model of multiple sclerosis, short-term (4 hour) preconditioning of human ASCs did not yield any therapeutic benefit and was correlated with worsened disease measures9. Thus, to improve translational potential, it is critically important to define the dynamic immunomodulatory response of MSCs and fine-tune these strategies to each pathological situation.”

  1. Also, the manuscript by Wise R.M. and colleagues is very similar to another one published by Wang et al. in 2017 (DOI 10.1186/s13287-017-0744-6) (cited ref n.53) in which it was demonstrated that inhibition of mTOR signaling using short-term rapamycin exposure enhanced the immunosuppressive functions of MSCs from bone marrow through a paracrine mechanism involving the COX-2 upregulation in the presence of IFN-γ and downstream PGE2. The Authors concluded that rapamycin could be a novel approach to improve the MSC-based therapeutic effect. To overcome this issue and to improve the quality of this manuscript, I suggest better emphasizing the originality of the work.

Response: We thank the reviewer for this suggestion, and to strengthen the emphasis on the novelty of our study, we have added the following to the Discussion:

“These unique temporal dynamics of COX2 and PGE2 have previously been shown in BMSCs, and have been suggested to result from activation of the Akt/Glycogen Synthase Kinase 3 Beta (GSK-3β) pathway rather than autophagy 3. Wang and colleagues demonstrated that short-term but not long-term, inhibition of mTORC1 with Rapamycin resulted in elevated COX2 gene expression, PGE2 secretion, and inhibition of proliferation in PBMCs. However, ASCs represent a more abundant and readily available source of stem cells compared to BMSCs and possess higher immunomodulatory potential4. Thus, the present study investigated whether ASCs demonstrate a similar temporally-distinct response to Rapamycin treatment to determine if this therapeutic strategy may be extended to a novel cell type.”

  1. The short-term autophagy preconditioning led to a strong upregulation of COX-2 gene expression. In bone marrow MSC it has been demonstrated that COX-1 expression was not affected by rapamycin (Wang et al. 2017 DOI 10.1186/s13287-017-0744-6). Does the COX-1 expression, which is constitutively expressed in most tissues and cells, is affected by rapamycin exposure?

Response: The reviewer raises an interesting question. According to a 2018 investigation by the group of Meilian Liu, mTORC1 is an important regulator of the COX2/PG signaling pathway in white adipose tissue and plays a role in stress response and adipose browning. They demonstrate that while manipulation of mTORC1 correlated well with COX2 expression and its downstream activity, COX1 remained constitutively expressed and relatively unaffected5. Additionally, while both COX1/2 isoforms do appear to play a role in MSC adipogenesis, only COX2 appears to be essential for immune response6; we, therefore, feel that a discussion about the COX1 isoform is beyond the scope of the present study.

Minor points:

- Statistical analysis: Data from independent experiments are presented as means ± standard deviations (SD). I believe it would be more accurate to use the standard error mean (SEM) instead of standard deviation.

Response: We agree with the reviewer and have updated the Methods section, analyses, and figure legends accordingly.

-Lines 117 and 170: Maybe I’m wrong, but I can’t see the Tables? Are they inserted in the supplementary materials or somewhere else?

Response: We thank the reviewer for pointing this out and apologize for the oversight. We have now included the data Tables.

-Line 142: It is not necessary or useful to insert the link of the image acquisition program.

Response: This has been removed.

- In line 199, authors presented the LC3-II western blot (WB) (Fig S1A) and reported this sentence “Western blot analysis revealed increased protein levels of LC3-II/β-Actin and decreased protein levels of p62/β-Actin, indicative of autophagy initiation (Fig. S1A, B)” but, the increase in the intensity of the LC3-II band is not absolutely evident. Moreover, the densitometric analysis, considering the SD values, shows no significant differences or a time-dependent growing trend. Can you replace the WB image of LC3-II? Moreover, do high rapamycin treatment times induce significant changes in p62 levels? I believe that these slight alterations of LC3-II and p62 expression are not to be considered to unmistakably confirm the autophagy induction.

Response: While we feel that the combination of our RT-qPCR and western blot data indicates induction of autophagy, we agree with the reviewer that this cannot be absolutely concluded from the data provided. We feel that this is partly due to the SD values and that greater sample sizes would potentially reach significance. However, we have now chosen a more representative blot for Supplementary figure 1 and updated the language in the Results and Discussion sections to reflect the presented data better:

Results: “In this study, hASCs were treated with Rapamycin (Rapa-ASCs) for 1, 4, 12, 24 and 48 hours, then examined for the expression of key autophagy genes and proteins. Transcriptional analysis of Rapa-ASCs demonstrated significant upregulation of the autophagy genes ATG7 and LC3B following prolonged, but not short-term exposure to Rapamycin (Fig. S1C), suggesting induction of autophagy. Western blot analysis revealed increased protein levels of LC3-II/β-Actin and decreased protein levels of p62/β-Actin; however, this failed to reach significance (Fig. S1A, B). To examine inhibition of autophagy, hASCs were treated with 3-MA (3MA-ASCs) for 1, 4, 12 and 24 hours, then probed for changes in autophagy gene and protein levels. Due to previous reports of autophagy induction with prolonged exposure to 3-MA[37], the longest time point (48-hours) was not included for this group. The 3MA-ASCs exhibited no change to LC3-II or p62 transcript or protein levels, indicating no induction of autophagosome formation (Fig. S2A, B).”

Discussion: “In the present study, the impacts of compounds known to either induce or inhibit autophagy on hASC immunomodulatory potential was assessed. Our data showed that 24-hour Rapa treatment increased the expression of the autophagy genes ATG7 and LC3B, while the expression of LC3-II protein was upregulated, albeit slightly suggestive of autophagic induction. Conversely, 3-MA, a class I and III PI3K inhibitor, obstructed autophagy initiation in hASCs. 3MA-ASCs displayed no significant change to autophagy transcripts or protein levels.”

- Supplementary Figures: Transcriptional analysis of ATG7 shall be included in the methods. Please, insert the legends of the supplementary figures.

Response: We apologize for this oversight and have included detailed figure legends with the supplementary figures. We have also included the following in the main text regarding ATG7 transcript levels:

“Furthermore, transcriptional analysis of Rapa-ASCs demonstrated significant upregulation of the autophagy genes ATG7 and LC3B following prolonged, but not short-term exposure to Rapamycin (Fig. S1C).”

-I really do not understand the choice of treatment times. Rapamycin is used for 1, 4, 12, 24, and 48 hours but not the 3MA (1, 4, 12, and 24 hours). The Authors should explain the reason.

Response: This is indeed an important point to address. This decision was based on reports of temporal differences in 3-MA inhibitory activity in the literature, showing that prolonged exposure induces autophagy7. The following has been added to the manuscript to provide a rationale for this decision:

Results: To examine inhibition of autophagy, hASCs were treated with 3-MA (3MA-ASCs) for 1, 4, 12 and 24 hours, then probed for changes in autophagy gene and protein levels. Due to previous reports of autophagy induction with prolonged exposure to 3-MA7, the longest time point (48-hours) was not included for this group. 

-Line 224: Correct (Fig. 1C) with (Fig. 1D).

Response: We apologize for this discrepancy and have corrected the manuscript by removing the incorrect reference to Fig. 1C.

-Lines 224-227: I suggest inserting images of the colonies.

Response: We have included images of cell colonies from the colony-forming units assay in Figures 1 D & E and have updated the figure legends.

“Figure 1: Autophagy preconditioning in hASCs does not alter self-renewal capacity or surface expression of MSC markers. (A) Representative images of hASCs morphology in culture after short and long-term exposure to autophagy preconditioning agents. Scale bar= 300 µm. Immunophenotype of Rapa-ASCs (B) and 3MA-ASCs (C) by surface proteins flow cytometric analysis. Data are presented as mean ± SEM of 3 independent experiments. Quantification and images of colony-forming units by Rapa-ASCs (D) and 3MA-ASCs (E). Data are presented as mean ± SEM of 4 independent experiments. All data comparing 3 groups are analyzed using one-way analysis of variance (ANOVA) with Tukey’s posthoc multiple comparisons, while CFU-F data are analyzed with unpaired student’s T-test. Abbreviations: Rapa, Rapamycin; 3-MA, 3-methyladenine; CFU-F, colony forming units-fibroblasts.”

- Figure 2: The COX-2 and relative PGE2 levels are presented. Is the COX-2 expression in the control cells (white square) totally absent? If so, how the evident level of PGE2 can be explained? Please justify.

Response: We thank the reviewer for this question and apologize for the confusion. In Figure 2B, relative quantification of COX2 transcript levels is provided using the ΔΔCt analysis method. In this case, the control samples are set at 1, and the fold-change of both Rapa-ASCs and 3MA-ASCs compared to untreated controls are presented. In Figure 2C, absolute quantification of secreted PGE2 is provided for all groups. To help clarify this, we have added the following to the figure legend:

“Figure 2: Autophagy preconditioning in hASCs alters expression of both anti-inflammatory and pro-inflammatory mediators. Rapa-ASCs (left) and 3MA-ASCs (right) were treated for 4 or 24 hours. Relative expression levels of anti-inflammatory (A) and pro-inflammatory (B) genes were measured via RT-qPCR using the ΔΔCt method. Data are presented as mean relative fold-change ± SD of 4 independent experiments. (C) Secreted PGE2 levels were measured via ELISA in 24-hour CM from control and autophagy preconditioned hASCs. Data are presented as means ± SD of 3 independent experiments. Statistical analysis was performed using one-way analysis of variance (ANOVA), and differences between the means are indicated with *p < 0.05, **p < 0.01, ***p < 0.001. Abbreviations: TGF-β, transforming growth factor-beta; IDO, indoleamine 2,3-dioxygenase; TSG-6, TNF Alpha Induced Protein 6, COX2, cyclooxygenase 2; IL-6, interleukin-6; IL-1β, interleukin-1 beta; PGE2, prostaglandin E2.

-Line 289: Please insert   ****p<0.0001

Response: This correction has been made.

Reviewer 2 Report

This paper is a very interesting research report proposing an improved method of culturing ADSCs for regenerative medicine. It is a novel approach to modulate inflammatory regulators by controlling autophagy with drugs to improve therapeutic effects. However, the description needs to be modified or added.

There is a table missing, so please add it.

Supplementary data should be added as regular figures.

The gene pathway from drug administration to regulatory factor secretion should be described. Please describe whether this pathway is specific to ADSCs or a general pathway.

I would like the authors to add a discussion of the specific diseases for which the prepared cells are considered to be effective, including the method of administration. Please also consider necessary animal experiments.

Author Response

We would like to thank all reviewers for their knowledgeable critique and insightful suggestions on how to improve our work entitled “Short-term Rapamycin Preconditioning Diminishes Therapeutic Efficacy of Human Adipose-Derived Stem Cells in a Murine Model of Multiple Sclerosis.” We have addressed all reviewer comments and concerns, and we feel they have improved the revised manuscript. All additional references have been incorporated into the manuscript. The methods have been clarified, and the Abstract, Introduction and Discussion sections have been strengthened due to the integrated edits. We have addressed the comments specific to each reviewer in the following sections.

REVIEWER 2

This paper is a very interesting research report proposing an improved method of culturing ADSCs for regenerative medicine. It is a novel approach to modulate inflammatory regulators by controlling autophagy with drugs to improve therapeutic effects. However, the description needs to be modified or added.

  1. There is a table missing, so please add it.

Response: We thank the reviewer for pointing this out and apologize for the oversight. We have now included the data Tables.

  1. Supplementary data should be added as regular figures.

Response: We appreciate the reviewer’s suggestion to include supplementary data in the main manuscript. However, because the most potent response--the upregulated COX2 transcripts and PGE2 secretion--was seen in both the Rapamycin and the 3-MA groups, we propose that the modulation be independent of the autophagy pathway. We, therefore, feel that this data is more suitable as supplementary information.

  1. The gene pathway from drug administration to regulatory factor secretion should be described. Please describe whether this pathway is specific to ADSCs or a general pathway.

Response: We agree that a description of the pathway would improve our Discussion. The following has been added to the Discussion section of the manuscript:

“mTOR, a serine/threonine kinase which forms the catalytic subunit of two distinct complexes mTORC1 and mTORC2, regulates a spectrum of cellular processes including cell growth, autophagy, cytoskeletal remodeling, proteostasis, and metabolism (elegantly reviewed here1). Rapamycin, an FDA-approved immunosuppressant, rapidly inhibits mTORC1 activity through interaction with the FRB (FKBP12/rapamycin-binding) domain2. Under physiological conditions, mTORC1 substrates negatively regulate mTORC2 kinase activity3. With Rapamycin inhibition of mTORC1, mTORC2 kinase activity is disinhibited, leading to phosphorylation of Akt at Ser473 and subsequently disrupts the enzyme glycogen synthase kinase 3 beta (GSK3B) blockade of the COX2 promoter region and activate COX2 transcription4–6. With prolonged Rapamycin exposure, mTORC2 is also inhibited, possibly explaining the rapid yet transient COX2 upregulation7. The temporally-defined actions of 3-MA may also explain the observed transcriptional changes. 3-MA persistently inhibits class I PI3K while inhibition of the class III PI3K, VPS34, is transient4. This was correlated with transient inhibition of autophagy and upregulated COX2 levels in human BMSCs with short-term exposure, but induction of autophagy and suppression of COX2 with long-term treatment. Thus, it may be that the selective inhibition of mTORC1 kinase activity, independent of the effects on autophagy, are responsible for the altered immunophenotype of ASCs.”

  1. I would like the authors to add a discussion of the specific diseases for which the prepared cells are considered to be effective, including the method of administration. Please also consider necessary animal experiments.

Response: We agree that adding this to our discussion will improve the manuscript’s impact. We have included the following in the Discussion section:

“Preconditioning strategies to improve immunomodulatory potency of MSCs for various anti-inflammatory and regenerative medicine applications is an active focus of the stem cell research field. Rapamycin has emerged as a promising candidate compound, and Rapa-treated MSCs from adipose8,9, bone marrow10, and umbilical cord11 have been examined in preclinical models. However, a wide range of dosage and exposure time has yielded varying success. It highlights the importance of defining the unique immunoregulatory consequences of these novel preconditioning strategies depending on experimental conditions (i.e. dose and exposure time), MSC tissue source, and therapeutic application. For example, short-term (2 hour) treatment of rat BMSCs improved survival and repair of damaged myocardium following transplant into an ischemia/reperfusion model10. However, in an animal model of multiple sclerosis, short-term (4 hour) preconditioning of human ASCs did not yield any therapeutic benefit and was correlated with worsened disease measures9. Thus, to improve translational potential, it is critically important to define the dynamic immunomodulatory response of MSCs and fine-tune these strategies to each pathological situation.”

“PGE2 is synthesized from arachidonic acid by the enzyme COX2 and has been proposed as an important mechanism contributing to the immunoregulatory actions of MSCs. In a mouse model of sepsis, concurrent IV administration of BMSCs pre-stimulated with LPS resulted in prolonged survival, which was correlated with elevated PGE2 secretion and modulation of host macrophage populations12. PGE2 has also been shown to be critical for MSCs’ ability to inhibit the proliferation of natural killer (NK) cells13, the maturation of dendritic cells (DC) from monocytes14,15, and the proliferation of PHA-stimulated T cells15.”

Reviewer 3 Report

Wise et al. provided a well written manuscript dealing with the question, whether preconditioning of human ASC has effects on their immunomodulatory potential upon IFNG stimulation. The authors provided data suggesting that short term activation of autophagy promotes the anti-inflammatory effect of ASC. Although the hypothesis evident, the study lacks convincing data that a) autophagy is really a prerequisite for the observed effects and b) the identified immunoregulatory cytokines are significant beyond qPCR expression data. Major and minor points are described below in detail.

Major points:

  • The mRNA differences between Rapamycin and 3-MA treated ASC are minor, at least in respect of the investigated proteins/cytokines. Figure 2 shows an about 2x increase of TSG6 in 4h Rapa treated vs. 3-MA treated ASC. This effect is reversed in 24h treated 3-MA cells. In addition, Rapamycin and 3-MA induced similar effects in respect to PGE2 levels before and after IFNG-treatment. These observations by their own suggest that downstream affected pathways might be responsible for immunomodulatory properties rather than induction of autophagy. To rule out that autophagy is indeed a prerequisite for the observed effects, the authors must provide convincing data according to accepted autophagy guidelines published in Klionsky et. al (Autophagy.2021Jan;17(1):1-382.doi: 10.1080/15548627.2020.1797280. Epub 2021 Feb 8).
  • In this respect the provided immunoblot data on LC3-II and p62 (supplement figure 1&2) are not correlating with data from quantification. For example, p62 levels clearly increase after 1 and 4 h Rapa treatment in the western blot image, but quantification describes a decrease. The authors must provide more convincing immunoblot data to support their quantification results.
  • In Figure 3 the important control 4h/24h Rapamycin or 3-MA and 24h without IFNG is missing. The authors have to provide these data. Importantly, these data would also provide insight, whether 4h/24h preconditioning induces a permanent (24h after precondition) or only transient effects on ASCs.
  • A functional assay testing the importance of the identified cytokine regulations is missing. The suggested immunomodulatory effects are concluded from qPCR data and literature only. The manuscript would definitively benefit from a functional assay, eg. altered immunoregulatory function/proliferation of PBMC or macrophages exposed to CM of preconditioned ASC. This functional proof will increase the impact of the paper significantly!!

Minor points:

Figure 1A: In my point of view, a change in morphology can not be derived from provided images. Please increase magnification and resolution of acquired images or use fluorescence markers (eg. cell membrane labelling).  

Figure 1B/C: Why the cells positive for CD45 although the cells have been purchased and completely characterized as stated in M&M?

Figure 1E/D: Please provide exemplified images of cell colonies after RAPA and 3-MA treatment. Are there any differences in size or morphology?

Line 244: Delete “preconditioning” since the cells were harvested directly after exposure to Rapa or 3-MA, thus it is just a treatment. If the term “preconditioning” is used, then the cells must be incubated for additional xy hours and then analysed. But this is not described in M&M or figure legend.

Line 329: For me it is not clear what is meant by the term “inflation”. I would rather use “induction”.

Line 331: The author shall also discuss increase of PGE2 in 24hr 3-MA preconditioned cells.  

Figure Suppl.1:The presented western blot data does not correlate with the quantification. For example, p62 levels clearly increase after 1 and 4 h Rapa treatment, but quantification describes a decrease. The authors must provide new immunoblots supporting their quantification results.

Figure Suppl.1&2: Figure legends are missing.

Author Response

We would like to thank all reviewers for their knowledgeable critique and insightful suggestions on how to improve our work entitled “Short-term Rapamycin Preconditioning Diminishes Therapeutic Efficacy of Human Adipose-Derived Stem Cells in a Murine Model of Multiple Sclerosis.” We have addressed all reviewer comments and concerns, and we feel they have improved the revised manuscript. All additional references have been incorporated into the manuscript. The methods have been clarified, and the Abstract, Introduction and Discussion sections have been strengthened due to the integrated edits. We have addressed the comments specific to each reviewer in the following sections.

REVIEWER 3

Wise et al. provided a well written manuscript dealing with the question, whether preconditioning of human ASC has effects on their immunomodulatory potential upon IFNG stimulation. The authors provided data suggesting that short term activation of autophagy promotes the anti-inflammatory effect of ASC. Although the hypothesis evident, the study lacks convincing data that a) autophagy is really a prerequisite for the observed effects and b) the identified immunoregulatory cytokines are significant beyond qPCR expression data. Major and minor points are described below in detail.

Major points:

  1. The mRNA differences between Rapamycin and 3-MA treated ASC are minor, at least in the investigated proteins/cytokines. Figure 2 shows an about 2x increase of TSG6 in 4h Rapa treated vs. 3-MA treated ASC. This effect is reversed in 24h treated 3-MA cells. In addition, Rapamycin and 3-MA induced similar effects on PGE2 levels before and after IFNG-treatment. These observations on their own suggest that downstream affected pathways might be responsible for immunomodulatory properties rather than induction of autophagy. To rule out that autophagy is indeed a prerequisite for the observed effects, the authors must provide convincing data according to accepted autophagy guidelines published in Klionsky et al. (Autophagy.2021Jan;17(1):1-382.doi: 10.1080/15548627.2020.1797280. Epub 2021 Feb 8). In this respect, the provided immunoblot data on LC3-II and p62 (supplement figure 1 & 2) are not correlating with data from quantification. For example, p62 levels clearly increase after 1 and 4 h Rapa treatment in the western blot image, but quantification describes a decrease. The authors must provide more convincing immunoblot data to support their quantification results.

Response: While we feel that the combination of our RT-qPCR and western blot data indicates induction of autophagy, we agree with the reviewer that this cannot be absolutely concluded from the data provided. We feel that this is partly due to the SD values and that greater sample sizes would potentially reach significance. However, we have now chosen a more representative blot for Supplementary figure 1 and updated the language in the Results and Discussion sections to reflect the presented data better:

Results: “In this study, hASCs were treated with Rapamycin (Rapa-ASCs) for 1, 4, 12, 24 and 48 hours, then examined for the expression of key autophagy genes and proteins. Transcriptional analysis of Rapa-ASCs demonstrated significant upregulation of the autophagy genes ATG7 and LC3B following prolonged, but not short-term exposure to Rapamycin (Fig. S1C), suggesting induction of autophagy. Western blot analysis revealed increased protein levels of LC3-II/β-Actin and decreased protein levels of p62/β-Actin; however, this failed to reach significance (Fig. S1A, B). To examine inhibition of autophagy, hASCs were treated with 3-MA (3MA-ASCs) for 1, 4, 12 and 24 hours, then probed for changes in autophagy gene and protein levels. Due to previous reports of autophagy induction with prolonged exposure to 3-MA [37], the longest time point (48-hours) was not included for this group. The 3MA-ASCs exhibited no change to LC3-II or p62 transcript or protein levels, indicating no induction of autophagosome formation (Fig. S2A, B).”

Discussion: “In the present study, the impacts of compounds known to either induce or inhibit autophagy on hASC immunomodulatory potential were assessed. Our data showed that 24-hour Rapa treatment increased the expression of the autophagy genes ATG7 and LC3B, while the expression of LC3-II protein was upregulated, albeit slightly suggestive of autophagic induction. Conversely, 3-MA, a class I and III PI3K inhibitor, obstructed autophagy initiation in hASCs. 3MA-ASCs displayed no significant change to autophagy transcripts or protein levels.” (Page #, Line #)

  1. In Figure 3 the important control 4h/24h Rapamycin or 3-MA and 24h without IFNG is missing. The authors have to provide these data. Importantly, these data would also provide insight, whether 4h/24h preconditioning induces a permanent (24h after precondition) or only transient effects on ASCs.

Response: While we agree with the reviewer that adding the controls without IFNG to figure 3 is essential, these data are presented in Figure 2. We feel that the critical metric is the comparison of the IFNγ-elicited immune response of pretreated versus control cells, as this is an estimate of how the cells would respond to an inflammatory post-transplant tissue niche.

  1. A functional assay testing the importance of the identified cytokine regulations is missing. The suggested immunomodulatory effects are concluded from qPCR data and literature only. The manuscript would definitively benefit from a functional assay, e.g. altered immunoregulatory function/proliferation of PBMC or macrophages exposed to CM of preconditioned ASC. This functional proof will increase the impact of the paper significantly!!

Response: We thank the reviewer for this suggestion, and we agree that additional experiments would enhance the impact of the study. We plan to pursue in-depth functional analyses of ASCs co-culture with immune cells and will be including these findings in future manuscripts. However, the data presented in this study (mRNA expression and ELISA) support our hypothesis that ASC immunomodulatory or therapeutic potential may be enhanced with short-term rapamycin.

Minor points:

-Figure 1A: In my point of view, a change in morphology cannot be derived from provided images. Please increase magnification and resolution of acquired images or use fluorescence markers (e.g. cell membrane labelling). 

Response: Although we agree with the reviewer that morphological changes can’t be detected from these images, unfortunately, we did not acquire a higher magnification at the experiment.

-Figure 1B/C: Why the cells positive for CD45 although the cells have been purchased and completely characterized as stated in M&M?

Response: ASCs are a heterogeneous mixture of cells expressing low levels of CD45 and CD31. Mesenchymal stem cells expressing CD45 have been reported as progenitor cells that have the ability of differentiation to blood vessels. In this study, we considered ASCs to be negative for CD45 as their expression was less than 10%. ­­­­

-Figure 1E/D: Please provide exemplified images of cell colonies after RAPA and 3-MA treatment. Are there any differences in size or morphology?

Response: Although the images are not taking at high resolution, they demonstrate no change in morphology or size. This is also supported by the ideas presented in Figure 1A.

-Line 244: Delete “preconditioning” since the cells were harvested directly after exposure to Rapa or 3-MA, thus it is just a treatment. If the term “preconditioning” is used, the cells must be incubated for additional xy hours and then analysed. But this is not described in M&M or figure legend.

Response: We agree with the reviewer and have changed the language in this section to reflect the experimental approach more accurately.

“To determine whether pharmacologically targeting the autophagy pathway alters the immune-modifying abilities of hASCs, cells were treated for either 4 or 24 hours with Rapa or 3-MA then analyzed with RT-qPCR for expression of common immune mediators. In Rapa-ASCs, 4-hour but not 24-hour treatment produced robust enhancement of TSG-6 mRNA (Fig. 2A). The opposite result was seen in 3MA-ASCs, with only the 24-hour group demonstrating enhanced expression of TGF-β and, to a lesser extent, TSG-6 (p=0.073). Further analysis revealed similar time-dependent differences in the expression of pro-inflammatory mediators (Fig. 2B). In both Rapa-ASCs and 3MA-ASCs, IL-6 was upregulated after 4 hours, but not 24 hours. Moreover, Rapa-ASCs, but not 3MA-ASCs, upregulated IL-1β after 4-hour treatment. Interestingly, both 4-hour Rapa-ASCs and 3MA-ASCs robustly upregulate COX2 transcription relative to control cells. This effect was not mirrored in secreted protein, as ELISA analysis of conditioned medium revealed elevation of the COX2 metabolite, PGE2, in 24-hour treated 3MA-ASCs only (Fig. 2C).”

-Line 329: For me it is not clear what is meant by the term “inflation”. I would rather use “induction”.

Response: We agree with the reviewer and have changed the language accordingly.

-Line 331: The author shall also discuss the increase of PGE2 in 24hr 3-MA preconditioned cells. 

Response:

We have added the following sentence to the discussion: “In this study, 4-hour preconditioned Rapa-ASCs and 3MA-ASCs both exhibited robust upregulation of COX2 gene expression. This induction was not seen in secreted PGE2, the protein substrate of COX2, with autophagy preconditioning alone in either 4-hour treatment group; however, PEG2 was significantly increased in ASCs treated with 3-MA for 24h.”

-Figure Suppl.1: The presented western blot data does not correlate with the quantification. For example, p62 levels clearly increase after 1 and 4 h Rapa treatment, but quantification describes a decrease. The authors must provide new immunoblots supporting their quantification results.

Response: This issue has been addressed under Major Point #2.

-Figure Suppl.1&2: Figure legends are missing.

Response: We apologize for this oversight and have added the relevant figure legends for the supplementary figures.

Reviewer 4 Report

In this manuscript by Wise et al, the authors aim to demonstrate the association between autophagy and immunomodulatory properties in human adipose-derived stem cells (hASCs). They discovered that treatment of hASCs with an autophagy stimulator rapamycin or an autophagy inhibitor 3-MA in the absence or presence of interferon-gamma resulted in various degrees of pro- and anti-inflammatory gene expression. They concluded that short term pre-conditioning of hASCs with rapamycin is a good strategy for promoting the immunomodulatory activity. While manipulation of the immune properties of hASCs as shown in this study gives great promise for its potential clinical application,

there are a few critical concerns that need to be addressed before this manuscript is in a publishable fashion. Specific comments are as follows:

1. In Figure S1A and B, the authors described "Western blot analysis revealed increased protein levels of LC3-II/β-Actin and decreased protein levels of p62/β-Actin, indicative of autophagy initiation". The p62 levels do not look very different with time, either in the representative (S1A) or the three original blots. The representative (S1A) even looks like there is an increase. The LC3 lipidation also seems inconsistent with the three repeats.

2. The observations that both stimulation and inhibition of autophagy activate certain genes in hASCs (e.g. TSG-6 and COX-2) need further discussion. Does it mean that the effects may be through an unidentified mechanism independent of autophagy or something compensatory from the treatment?

3. There is some misinterpretation of the data in Figure 3. For example, in Figure 3A, the authors described "Analysis of anti-inflammatory cytokines revealed that 24-hour preconditioned Rapa-ASCs further enhanced expression of both TGF-β and TSG-6 transcripts over IFNγ-treated controls". These cannot be observed in the figure. The authors also described "At both time points, Rapa-ASCs demonstrated further upregulation of the pro-inflammatory mediators COX2, IL-6, and TNFα mRNA, while 3MA-ASCs only exhibited increased expression of COX2 and TNFα with 4-hour precondition-274 ing (Figure 3B)". Again, these are not oberved in the figure, either. In both figures, statistical comparisons are only made with the pre-conditioned hASCs but not interferon-gamma treated hASCs.

Author Response

We would like to thank all reviewers for their knowledgeable critique and insightful suggestions on how to improve our work entitled “Short-term Rapamycin Preconditioning Diminishes Therapeutic Efficacy of Human Adipose-Derived Stem Cells in a Murine Model of Multiple Sclerosis.” We have addressed all reviewer comments and concerns, and we feel they have improved the revised manuscript. All additional references have been incorporated into the manuscript. The methods have been clarified, and the Abstract, Introduction and Discussion sections have been strengthened due to the integrated edits. We have addressed the comments specific to each reviewer in the following sections.

REVIEWER 4

In this manuscript by Wise et al, the authors aim to demonstrate the association between autophagy and immunomodulatory properties in human adipose-derived stem cells (hASCs). They discovered that treatment of hASCs with an autophagy stimulator rapamycin or an autophagy inhibitor 3-MA in the absence or presence of interferon-gamma resulted in various degrees of pro- and anti-inflammatory gene expression. They concluded that short term pre-conditioning of hASCs with rapamycin is a good strategy for promoting the immunomodulatory activity. While manipulation of the immune properties of hASCs as shown in this study gives great promise for its potential clinical application, there are a few critical concerns that need to be addressed before this manuscript is in a publishable fashion. Specific comments are as follows:

  1. In Figure S1A and B, the authors described “Western blot analysis revealed increased protein levels of LC3-II/β-Actin and decreased protein levels of p62/β-Actin, indicative of autophagy initiation”. The p62 levels do not look very different with time, either in the representative (S1A) or the three original blots. The representative (S1A) even looks like there is an increase. The LC3 lipidation also seems inconsistent with the three repeats.

Response: While we feel that the combination of our RT-qPCR and western blot data indicates induction of autophagy, we agree with the reviewer that this cannot be absolutely concluded from the data provided. We feel that this is partly due to the SD values and that greater sample sizes would potentially reach significance. However, we have now chosen a more representative blot for Supplementary Figure 1 and updated the language in the Results and Discussion sections to reflect the presented data better:

Results: “In this study, hASCs were treated with Rapamycin (Rapa-ASCs) for 1, 4, 12, 24 and 48 hours, then examined for the expression of key autophagy genes and proteins. Transcriptional analysis of Rapa-ASCs demonstrated significant upregulation of the autophagy genes ATG7 and LC3B following prolonged, but not short-term exposure to Rapamycin (Fig. S1C), suggesting induction of autophagy. Western blot analysis revealed increased protein levels of LC3-II/β-Actin and decreased protein levels of p62/β-Actin; however, this failed to reach significance (Fig. S1A, B). To examine inhibition of autophagy, hASCs were treated with 3-MA (3MA-ASCs) for 1, 4, 12 and 24 hours, then probed for changes in autophagy gene and protein levels. Due to previous reports of autophagy induction with prolonged exposure to 3-MA [37], the longest time point (48-hours) was not included for this group. The 3MA-ASCs exhibited no change to LC3-II or p62 transcript or protein levels, indicating no induction of autophagosome formation (Fig. S2A, B).”

Discussion: “In the present study, the impacts of compounds known to either induce or inhibit autophagy on hASC immunomodulatory potential were assessed. Our data showed that 24-hour Rapa treatment increased the expression of the autophagy genes ATG7 and LC3B, while the expression of LC3-II protein was upregulated, albeit slightly suggestive of autophagic induction. Conversely, 3-MA, a class I and III PI3K inhibitor, obstructed autophagy initiation in hASCs. 3MA-ASCs displayed no significant change to autophagy transcripts or protein levels.”

  1. The observations that both stimulation and inhibition of autophagy activate certain genes in hASCs (e.g. TSG-6 and COX-2) need further discussion. Does it mean that the effects may be through an unidentified mechanism independent of autophagy or something compensatory from the treatment?

Response: We agree with the observation that some changes to immunomodulatory behavior may be the result of autophagy-independent mechanisms or the temporally-defined actions of both Rapamycin and 3-MA. We have therefore added this to the Discussion as potential alternative explanations to altered hASC anti-inflammatory potential:

“These unique temporal dynamics of COX2 and PGE2 have previously been shown in BMSCs, and have been suggested to result from activation of the Akt/Glycogen Synthase Kinase 3 Beta (GSK-3β) pathway rather than autophagy [54]. mTOR, a serine/threonine kinase that forms the catalytic subunit of two distinct complexes, mTORC1 and mTORC2, regulates a spectrum of cellular processes, including cell growth, autophagy, cytoskeletal remodeling, proteostasis, and metabolism (elegantly reviewed here1). Rapamycin, an FDA-approved immunosuppressant, rapidly inhibits mTORC1 activity through interaction with the FRB (FKBP12/rapamycin-binding) domain2. Under physiological conditions, mTORC1 substrates negatively regulate mTORC2 kinase activity3. With Rapamycin inhibition of mTORC1, mTORC2 kinase activity is disinhibited, leading to phosphorylation of Akt at Ser473 and subsequently disrupts the enzyme glycogen synthase kinase 3 beta (GSK3B) blockade of the COX2 promoter region and activate COX2 transcription4–6. With prolonged Rapamycin exposure, mTORC2 is also inhibited, possibly explaining the rapid yet transient COX2 upregulation7. The temporally-defined actions of 3-MA may also explain the observed transcriptional changes. 3-MA persistently inhibits class I PI3K while inhibition of the class III PI3K, VPS34, is transient4. This was correlated with transient inhibition of autophagy and upregulated COX2 levels in human BMSCs with short-term exposure, but induction of autophagy and suppression of COX2 with long-term treatment. Thus, it may be that the selective inhibition of mTORC1 kinase activity, independent of the effects on autophagy, is responsible for the altered immunophenotype of ASCs.

Wang and colleagues demonstrated that short-term but not long-term, inhibition of mTORC1 with Rapamycin resulted in elevated COX2 gene expression, PGE2 secretion, and inhibition of proliferation in PBMCs. However, ASCs represent a more abundant and readily available source of stem cells than BMSCs and possess higher immunomodulatory potential [3]. Thus, the present study investigated whether ASCs demonstrate a similar temporally-distinct response to Rapamycin treatment to determine if this therapeutic strategy may be extended to a novel cell type.”

  1. There is some misinterpretation of the data in Figure 3. For example, in Figure 3A, the authors described “Analysis of anti-inflammatory cytokines revealed that 24-hour preconditioned Rapa-ASCs further enhanced expression of both TGF-β and TSG-6 transcripts over IFNγ-treated controls”. These cannot be observed in the figure. The authors also described “At both time points, Rapa-ASCs demonstrated further upregulation of the pro-inflammatory mediators COX2, IL-6, and TNFα mRNA, while 3MA-ASCs only exhibited increased expression of COX2 and TNFα with 4-hour preconditioning (Figure 3B)”. Again, these are not observed in the figure, either. In both figures, statistical comparisons are only made with the pre-conditioned hASCs but not interferon-gamma treated hASCs.

Response:

We thank the reviewer for this comment and have clarified the analysis and interpretation of the data in Figure 3. While Figure 2 compares the transcriptional and secretory response of control versus pretreated hASCs, Figure 3 compares all IFNγ-treated groups (control, Rapa-, or 3-MA-pre-treated) with the baseline untreated control group which received neither a pre-treatment nor the subsequent IFNγ stimulation. This analysis compares how pretreated hASCs may respond to an inflammatory tissue niche in contrast to control hASCs. In addition, we have clarified our language to represent the data presented in the Results section more accurately:

  • Autophagy preconditioning of hASCs alters the response to pro-inflammatory stimulation with hIFNγ

To investigate the effects of autophagy preconditioning on hASC response to inflammatory signals, cells were pretreated for either 4 or 24 hours followed by 24-hour stimulation with the classic pro-inflammatory cytokine hIFNγ. Analysis of anti-inflammatory cytokines revealed that 4-hour preconditioned Rapa-ASCs significantly upregulated the expression of TGF-β. In contrast, 24-hour preconditioned Rapa-ASCs greatly enhanced expression of TGF-β, IDO and TSG-6 transcripts in response to hIFNγ-stimulation (Fig. 3A). In contrast, control hASCs with neither Rapa nor 3-MA pre-treatment demonstrated enhanced levels of IDO transcripts only following hIFNγ-stimulation. Furthermore, while prolonged preconditioning with 3-MA did not elicit elevation of anti-inflammatory cytokine transcripts, 4-hour 3-MA did induce a significant increase of IDO transcripts (Fig. 3A). Examination of pro-inflammatory cytokines revealed that control hASCs stimulated with hIFNγ increased expression of COX2 transcripts (Fig. 3B). However, with 4-hour Rapa preconditioning, hASCs upregulated COX2 and IL-6 transcripts, while 24-hour Rapa-ASCs augmented COX2, IL-6 and IL-1β mRNA (Fig. 3B). Similarly, 3MA-ASCs increased COX2, IL-6 and IL-1β transcripts expression with 4-hour preconditioning, but no differences from control hASCs with 24-hour preconditioning. Consistent with transcriptional upregulation of COX2 in hIFNγ-stimulated Rapa-ASCs and 3MA-ASCs, secreted PGE2 protein was significantly higher in both groups following short-term, but not long-term, autophagy preconditioning compared to both untreated controls and those stimulated with hIFNγ (Fig. 3C).

Round 2

Reviewer 1 Report

The authors have revised the manuscript in light of suggestions and critical issues. In my opinion, the manuscript in the revised form is now suitable for publication in Cells.

Author Response

We thank the reviewer for their suggestions in improving this manuscript and their support for publication.

Reviewer 3 Report

Authors responded to most comments. It can be accepted.

Author Response

(The authors gave the same response as above.)

Reviewer 4 Report

In the revised version of  the manuscript by Wise et al, the authors answered some of the questions by the reviewer. However, the critical concerns still remain and are listed as follows:

1. The major question still exists. Both the activator and inhibitor of autophagy up-regulate the genes of interest, maybe with a different time frame. Does that mean the up-regulation may have nothing to do with autophagy? What mechanism makes this common effects? How does this observation benefit future applications?

2. Autophagy preconditioning does not alter transcriptional response to pro-inflammatory stimulation with hIFNγ as stated in Figure 3A and B. IFN treatment changes the level of genes examined compared to the untreated control but pre-conditioning does not make any further changes in nearly all cases.

Author Response

Reviewer 4

In the revised version of the manuscript by Wise et al, the authors answered some of the questions by the reviewer. However, the critical concerns still remain and are listed as follows:

  1. The major question still exists. Both the activator and inhibitor of autophagy up-regulate the genes of interest, maybe with a different time frame. Does that mean the up-regulation may have nothing to do with autophagy? What mechanism makes this common effects? How does this observation benefit future applications?

Response: We thank the reviewer for highlighting this very important aspect of our study, and agree that this finding should be better emphasized and put into the larger context. We have therefore expanded our Results, Discussion, and Conclusion sections to address the strong evidence suggesting that the robust secretory alterations seen in our pre-treated, IFNγ-stimulated ASCs may be due to mechanisms entirely outside of the autophagy pathway. The following passages have been added/amended based on this feedback:

Results section 3.4: “Taken together, these findings show that while transcriptional activity is altered based on unique temporal and compound-dependent patterns, the robust upregulation of PGE2 release in both acute Rapa and 3-MA preconditioning strategies suggest the involvement of an autophagy-independent mechanism that is currently unknown.”

Discussion: “We also demonstrated that both compounds had similar impact on secretory activity of ASCs in response to inflammatory activation, which strongly suggests a shared molecular mechanism that is independent of their actions on autophagy.”

Conclusion: “Drawing on evidence from bone marrow-derived stem cells (BMSCs), we find that compounds commonly used to modify the autophagy pathway result in “primed” ASCs capable of producing higher levels of immunomodulatory genes and proteins.  We demonstrate that PGE2, a key contributing factor to stem cell therapeutic efficacy in multiple sclerosis (MS), sepsis, inflammatory bowel disease (IBD), and arthritis, was highly upregulated in ASCs preconditioned with either short-term Rapamycin or 3-MA, indicating that the immunomodulatory affects of these compounds may, in fact, derive from mechanisms of action beyond their impact on autophagy. We propose that this represents a promising strategy for enhancing the therapeutic potential and possibly the translational success of ASCs for inflammation-driven disease states that warrants further investigation and delination of mechanisms involved.

We have also included the further reference of the unique, GSK3β-dependent mechanisms of 3MA, which may explain the similar enhancement of PGE2 secretion with Rapamycin in the short-term treatment groups.

Reference #57: Lin, Y.-C.; Kuo, H.-C.; Wang, J.-S.; Lin, W.-W. Regulation of Inflammatory Response by 3-Methyladenine Involves the Coordinative Actions on Akt and Glycogen Synthase Kinase 3β Rather than Autophagy. J. Immunol. 2012, 189 (8), 4154–4164. https://doi.org/10.4049/JIMMUNOL.1102739.

  1. Autophagy preconditioning does not alter transcriptional response to pro-inflammatory stimulation with hIFNγ as stated in Figure 3A and B. IFNγ treatment changes the level of genes examined compared to the untreated control but pre-conditioning does not make any further changes in nearly all cases.

Response: In figure 3A and 3B, we analyzed the fold-change in transcript levels of all IFNγ-stimulated groups (control, Rapa-pretreated, and 3MA-pretreated) in comparison to a single, unstimulated control group. We chose this analytical approach to examine as closely as possible the transcriptional behavior of our cells following transplant, and thus estimate how either Rapa or 3-MA pretreatment altered their to potential to modulate inflammation in an inflammatory microenvironment. We have clarified our results section to better reflect our analytical approach, and have specified that all data are relative to unstimulated control ASCs. The following has been added to Results section 3.4:

Results section 3.4: “To investigate the effects of autophagy preconditioning on hASC immunomodulatory response to signals they may encounter following administration into an inflammatory microenvironment, cells were pre-treated for either 4- or 24-hours followed by 24-hour stimulation with the classic pro-inflammatory cytokine hIFNγ. Our findings demonstrate that, in comparison to unstimulated, non-pretreated control ASCs, 4-hour Rapa-preconditioning resulted in significant elevation of TGF-β, COX2, and IL-6, while 24-hour Rapa-preconditioning resulted in upregulation of TGF-β, IDO, TSG-6, and COX2, as well as the inflammatory cytokines IL-6 and IL-1β. We also found that 4-hour 3MA-preconditioning resulted in upregulation of IDO, COX2, IL-6, and IL-1β; while 24-hour 3MA-preconditioning only increased expression of COX2 transcripts (Fig. 3A, B). This contrasted with our non-preconditioned control ASCs, which in response to hIFNγ stimulation only displayed significant upregulation of IDO and COX2 transcripts when compared to non- IFNγ-stimulated ASCs. Perhaps most interesting, we demonstrated that secretion of PGE2 was upregulated by both 4-hour Rapa and 4-hour 3MA, which was significantly higher than both unstimulated control and IFNγ-stimulated control cells, as well as both 24-hour preconditioned groups (Fig. 3C).”